# APPROXIMATE VANISHING IDEAL COMPUTATIONS AT SCALE

**Elias Wirth**
Institute of Mathematics
Berlin Institute of Technology
Berlin, Germany
wirth@math.tu-berlin.de

**Hiroshi Kera**
Graduate School of Engineering
Chiba University
Chiba, Japan
kera.hiroshi@gmail.com

**Sebastian Pokutta**
Institute of Mathematics & AI in Society, Science, and Technology
Berlin Institute of Technology & Zuse Institute Berlin
Berlin, Germany
pokutta@zib.de

## ABSTRACT

The *vanishing ideal* of a set of points $X = \{\mathbf{x}_1, \ldots, \mathbf{x}_m\} \subseteq \mathbb{R}^n$ is the set of polynomials that evaluate to $0$ over all points $\mathbf{x} \in X$ and admits an efficient representation by a finite subset of generators. In practice, to accommodate noise in the data, algorithms that construct generators of the *approximate vanishing ideal* are widely studied but their computational complexities remain expensive. In this paper, we scale up the *oracle approximate vanishing ideal algorithm* (OAVI), the only generator-constructing algorithm with known learning guarantees. We prove that the computational complexity of OAVI is not superlinear, as previously claimed, but linear in the number of samples $m$. In addition, we propose two modifications that accelerate OAVI's training time: Our analysis reveals that replacing the *pairwise conditional gradients algorithm*, one of the solvers used in OAVI, with the faster *blended pairwise conditional gradients algorithm* leads to an exponential speed-up in the number of features $n$. Finally, using a new *inverse Hessian boosting* approach, intermediate convex optimization problems can be solved almost instantly, improving OAVI's training time by multiple orders of magnitude in a variety of numerical experiments.

## 1 INTRODUCTION

High-quality features are essential for the success of machine-learning algorithms (Guyon & Elisseeff, 2003) and as a consequence, feature transformation and selection algorithms are an important area of research (Kusiak, 2001; Van Der Maaten et al., 2009; Abdi & Williams, 2010; Paul et al., 2021; Manikandan & Abirami, 2021; Carderera et al., 2021). A recently popularized technique for extracting nonlinear features from data is the concept of the vanishing ideal (Heldt et al., 2009; Livni et al., 2013), which lies at the intersection of machine learning and computer algebra. Unlike conventional machine learning, which relies on a manifold assumption, vanishing ideal computations are based on an algebraic set[1] assumption, for which powerful theoretical guarantees are known (Vidal et al., 2005; Livni et al., 2013; Globerson et al., 2017). The core concept of vanishing ideal computations is that any data set $X = \{\mathbf{x}_1, \ldots, \mathbf{x}_m\} \subseteq \mathbb{R}^n$ can be described by its *vanishing ideal*, $\mathcal{I}_X = \{g \in \mathcal{P} \mid g(\mathbf{x}) = 0 \text{ for all } \mathbf{x} \in X\}$, where $\mathcal{P}$ is the polynomial ring over $\mathbb{R}$ in $n$ variables. Despite $\mathcal{I}_X$ containing infinitely many polynomials, there exists a finite number of *generators* of $\mathcal{I}_X$, $g_1, \ldots, g_k \in \mathcal{I}_X$ with $k \in \mathbb{N}$, such that any polynomial $h \in \mathcal{I}_X$ can be written as $h = \sum_{i=1}^{k} g_i h_i$, where $h_i \in \mathcal{P}$ for all $i \in \{1, \ldots, k\}$ (Cox et al., 2013). Thus, the generators share any sample $\mathbf{x} \in X$ as a common root, capture the nonlinear structure of the data, and succinctly represent the vanishing ideal $\mathcal{I}_X$. Due to noise in empirical data, we are interested in constructing generators of

---

[1]A set $X \subseteq \mathbb{R}^n$ is *algebraic* if it is the set of common roots of a finite set of polynomials.

the *approximate vanishing ideal*, the ideal generated by the set of polynomials that approximately evaluate to 0 for all $\mathbf{x} \in X$ and whose leading term coefficient is 1, see Definition 2.2. For classification tasks, constructed generators can, for example, be used to transform the features of the data set $X \subseteq \mathbb{R}^n$ such that the data becomes linearly separable (Livni et al., 2013) and training a linear kernel *support vector machine* (SVM) (Suykens & Vandewalle, 1999) on the feature-transformed data results in excellent classification accuracy. Various algorithms for the construction of generators of the approximate vanishing ideal exist (Heldt et al., 2009; Fassino, 2010; Limbeck, 2013; Livni et al., 2013; Iraji & Chitsaz, 2017; Kera & Hasegawa, 2020; Kera, 2022), but among them, the *oracle approximate vanishing ideal algorithm* (OAVI) (Wirth & Pokutta, 2022) is the only one capable of constructing sparse generators and admitting learning guarantees. More specifically, CGAVI (OAVI with *Frank-Wolfe algorithms* (Frank & Wolfe, 1956), a.k.a. *conditional gradients algorithms* (CG) (Levitin & Polyak, 1966) as a solver) exploits the sparsity-inducing properties of CG to construct sparse generators and, thus, a robust and interpretable corresponding feature transformation. Furthermore, generators constructed with CGAVI vanish on out-sample data and the combined approach of transforming features with CGAVI for a subsequently applied linear kernel SVM inherits the margin bound of the SVM (Wirth & Pokutta, 2022). Despite OAVI's various appealing properties, the computational complexities of vanishing ideal algorithms for the construction of generators of the approximate vanishing ideal are superlinear in the number of samples $m$. With training times that increase at least cubically with $m$, vanishing ideal algorithms have yet to be applied to large-scale machine-learning problems.

## 1.1 CONTRIBUTIONS

In this paper, we improve and study the scalability of OAVI.

**Linear computational complexity in $m$.** Up until now, the analysis of computational complexities of approximate vanishing ideal algorithms assumed that generators need to vanish exactly, which gave an overly pessimistic estimation of the computational cost. For OAVI, we exploit that generators only have to vanish approximately and prove that the computational complexity of OAVI is not superlinear but linear in the number of samples $m$ and polynomial in the number of features $n$.

**Solver improvements.** OAVI repeatedly calls a solver of quadratic convex optimization problems to construct generators. By replacing the *pairwise conditional gradients algorithm* (PCG) (Lacoste-Julien & Jaggi, 2015) with the faster *blended pairwise conditional gradients algorithm* (BPCG) (Tsuji et al., 2022), we improve the dependence of the time complexity of OAVI on the number of features $n$ in the data set by an exponential factor.

**Inverse Hessian boosting (IHB).** OAVI solves a series of quadratic convex optimization problems that differ only slightly and we can efficiently maintain and update the inverse of the corresponding Hessians. *Inverse Hessian boosting* (IHB) then refers to the procedure of passing a starting vector, computed with inverse Hessian information, close to the optimal solution to the convex solver used in OAVI. Empirically, IHB speeds up the training time of OAVI by multiple orders of magnitude.

**Large-scale numerical experiments.** We perform numerical experiments on data sets of up to two million samples, highlighting that OAVI is an excellent large-scale feature transformation method.

## 1.2 RELATED WORK

The *Buchberger-Möller algorithm* was the first method for constructing generators of the vanishing ideal (Möller & Buchberger, 1982). Its high susceptibility to noise was addressed by Heldt et al. (2009) with the *approximate vanishing ideal algorithm* (AVI), see also Fassino (2010); Limbeck (2013). The latter introduced two algorithms that construct generators term by term instead of degree-wise such as AVI, the *approximate Buchberger-Möller algorithm* (ABM) and the *border bases approximate Buchberger-Möller algorithm*. The aforementioned algorithms are monomial-aware, that is, they require an explicit ordering of terms and construct generators as linear combinations of monomials. However, monomial-awareness is an unattractive property: Changing the order of the features changes the outputs of the algorithms. Monomial-agnostic approaches such as *vanishing component analysis* (VCA) (Livni et al., 2013) do not suffer from this shortcoming, as they construct generators as linear combinations of polynomials. VCA found success in hand posture recognition, solution selection using genetic programming, principal variety analysis for nonlinear data modeling, and independent signal estimation for blind source separation (Zhao & Song, 2014; Kera & Iba, 2016; Iraji & Chitsaz, 2017; Wang & Ohtsuki, 2018). The disadvantage of foregoing the term ordering is that VCA sometimes constructs multiple orders of magnitude more generators

than monomial-aware algorithms (Wirth & Pokutta, 2022). Furthermore, VCA is susceptible to the *spurious vanishing problem*: Polynomials that do not capture the nonlinear structure of the data but whose coefficient vector entries are small become generators, and, conversely, polynomials that capture the data well but whose coefficient vector entries are large get treated as non-vanishing. The problem was partially addressed by Kera & Hasegawa (2019; 2020; 2021).

## 2 PRELIMINARIES

Throughout, let $\ell, k, m, n \in \mathbb{N}$. We denote vectors in bold and let $\mathbf{0} \in \mathbb{R}^n$ denote the 0-vector. Sets of polynomials are denoted by capital calligraphic letters. We denote the set of terms (or monomials) and the polynomial ring over $\mathbb{R}$ in $n$ variables by $\mathcal{T}$ and $\mathcal{P}$, respectively. For $\tau \geq 0$, a polynomial $g = \sum_{i=1}^{k} c_i t_i \in \mathcal{P}$ with $\mathbf{c} = (c_1, \ldots, c_k)^\mathsf{T}$ is said to be $\tau$-*bounded* in the $\ell_1$-norm if the $\ell_1$-norm of its coefficient vector is bounded by $\tau$, that is, if $\|g\|_1 := \|\mathbf{c}\|_1 \leq \tau$. Given a polynomial $g \in \mathcal{P}$, let $\deg(g)$ denote its *degree*. The sets of polynomials in $n$ variables of and up to degree $d \in \mathbb{N}$ are denoted by $\mathcal{P}_d$ and $\mathcal{P}_{\leq d}$, respectively. Similarly, for a set of polynomials $\mathcal{G} \subseteq \mathcal{P}$, let $\mathcal{G}_d = \mathcal{G} \cap \mathcal{P}_d$ and $\mathcal{G}_{\leq d} = \mathcal{G} \cap \mathcal{P}_{\leq d}$. We often assume that $X = \{\mathbf{x}_1, \ldots, \mathbf{x}_m\} \subseteq [0,1]^n$, a form that can be realized, for example, via *min-max feature scaling*. Given a polynomial $g \in \mathcal{P}$ and a set of polynomials $\mathcal{G} = \{g_1, \ldots, g_k\} \subseteq \mathcal{P}$, define the *evaluation vector* of $g$ and *evaluation matrix* of $\mathcal{G}$ over $X$ as $g(X) = (g(\mathbf{x}_1), \ldots, g(\mathbf{x}_m))^\mathsf{T} \in \mathbb{R}^m$ and $\mathcal{G}(X) = (g_1(X), \ldots, g_k(X)) \in \mathbb{R}^{m \times k}$, respectively. Further, define the *mean squared error* of $g$ over $X$ as $\mathrm{mse}(g, X) = \frac{1}{|X|} \|g(X)\|_2^2 = \frac{1}{m} \|g(X)\|_2^2$.

OAVI sequentially processes terms according to a so-called *term ordering*, as is necessary for any monomial-aware algorithm. For ease of presentation, we restrict our analysis to the *degree-lexicographical ordering of terms* (DegLex) (Cox et al., 2013), denoted by $<_\sigma$. For example, given the terms $t, u, v \in \mathcal{T}_1$, DegLex works as follows: $\mathbb{1} <_\sigma t <_\sigma u <_\sigma v <_\sigma t^2 <_\sigma t \cdot u <_\sigma t \cdot v <_\sigma u^2 <_\sigma u \cdot v <_\sigma v^2 <_\sigma t^3 <_\sigma \ldots$, where $\mathbb{1}$ denotes the constant-1 term. Given a set of terms $\mathcal{O} = \{t_1, \ldots, t_k\}_\sigma \subseteq \mathcal{T}$, the subscript $\sigma$ indicates that $t_1 <_\sigma \ldots <_\sigma t_k$.

**Definition 2.1** (Leading term (coefficient)). Let $g = \sum_{i=1}^{k} c_i t_i \in \mathcal{P}$ with $c_i \in \mathbb{R}$ and $t_i \in \mathcal{T}$ for all $i \in \{1, \ldots, k\}$ and let $j \in \{1, \ldots, k\}$ such that $t_j >_\sigma t_i$ for all $i \in \{1, \ldots, k\} \setminus \{j\}$. Then, $t_j$ and $c_j$ are called *leading term* and *leading term coefficient* of $g$, denoted by $\mathrm{lt}(g) = t_j$ and $\mathrm{ltc}(g) = c_j$, respectively.

We thus define approximately vanishing polynomials via the mean squared error as follows.

**Definition 2.2** (Approximately vanishing polynomial). Let $X = \{\mathbf{x}_1, \ldots, \mathbf{x}_m\} \subseteq \mathbb{R}^n$, $\psi \geq 0$, and $\tau \geq 2$. A polynomial $g \in \mathcal{P}$ is $\psi$-*approximately vanishing* (over $X$) if $\mathrm{mse}(g, X) \leq \psi$. If also $\mathrm{ltc}(g) = 1$ and $\|g\|_1 \leq \tau$, then $g$ is called $(\psi, 1, \tau)$-*approximately vanishing* (over $X$).

In the definition above, we fix the leading term coefficient of polynomials to address the spurious vanishing problem, and the requirement that polynomials are $\tau$-bounded in the $\ell_1$-norm is necessary for the learning guarantees of OAVI to hold.

**Definition 2.3** (Approximate vanishing ideal). Let $X = \{\mathbf{x}_1, \ldots, \mathbf{x}_m\} \subseteq \mathbb{R}^n$, $\psi \geq 0$, and $\tau \geq 2$. The $(\psi, \tau)$-*approximate vanishing ideal* (over $X$), $\mathcal{I}_X^{\psi, \tau}$, is the ideal generated by all $(\psi, 1, \tau)$-approximately vanishing polynomials over $X$.

For $\psi = 0$ and $\tau = \infty$, it holds that $\mathcal{I}_X^{0, \infty} = \mathcal{I}_X$, that is, the approximate vanishing ideal becomes the vanishing ideal. Finally, we introduce the generator-construction problem addressed by OAVI.

**Problem 2.4** (Setting). *Let $X = \{\mathbf{x}_1, \ldots, \mathbf{x}_m\} \subseteq \mathbb{R}^n$, $\psi \geq 0$, and $\tau \geq 2$. Construct a set of $(\psi, 1, \tau)$-approximately vanishing generators of $\mathcal{I}_X^{\psi, \tau}$.*

Recall that for $t, u \in \mathcal{T}$, $t$ *divides (or is a divisor of) $u$*, denoted by $t \mid u$, if there exists $v \in \mathcal{T}$ such that $t \cdot v = u$. If $t$ does not divide $u$, we write $t \nmid u$. OAVI constructs generators of the approximate vanishing ideal of degree $d \in \mathbb{N}$ by checking whether terms of degree $d$ are leading terms of an approximately vanishing generator. As explained in Wirth & Pokutta (2022), OAVI does not have to consider all terms of degree $d$ but only those contained in the subset defined below.

**Definition 2.5** (Border). Let $\mathcal{O} \subseteq \mathcal{T}$. The *(degree-$d$) border* of $\mathcal{O}$ is defined as $\partial_d \mathcal{O} = \{u \in \mathcal{T}_d : t \in \mathcal{O}_{\leq d-1} \text{ for all } t \in \mathcal{T}_{\leq d-1} \text{ such that } t \mid u\}$.

---

**Algorithm 1:** Oracle approximate vanishing ideal algorithm (OAVI)

---

**Input** : A data set $X = \{\mathbf{x}_1, \ldots, \mathbf{x}_m\} \subseteq \mathbb{R}^n$ and parameters $\psi \geq 0$ and $\tau \geq 2$.
**Output:** A set of polynomials $\mathcal{G} \subseteq \mathcal{P}$ and a set of monomials $\mathcal{O} \subseteq \mathcal{T}$.

---

1   $d \leftarrow 1, \mathcal{O} = \{t_1\}_\sigma \leftarrow \{\mathbb{1}\}_\sigma, \mathcal{G} \leftarrow \emptyset$
2   **while** $\partial_d \mathcal{O} = \{u_1, \ldots, u_k\}_\sigma \neq \emptyset$ **do**
3     **for** $i = 1, \ldots, k$ **do**
4       $\ell \leftarrow |\mathcal{O}| \in \mathbb{N}, A \leftarrow \mathcal{O}(X) \in \mathbb{R}^{m \times \ell}, \mathbf{b} \leftarrow u_i(X) \in \mathbb{R}^m, P = \{\mathbf{y} \in \mathbb{R}^\ell \mid \|\mathbf{y}\|_1 \leq \tau - 1\}$
5       $\mathbf{c} \in \operatorname{argmin}_{\mathbf{y} \in P} \frac{1}{m} \|A\mathbf{y} + \mathbf{b}\|_2^2$    ▷ call to convex optimization oracle
6       $g \leftarrow \sum_{j=1}^\ell c_j t_j + u_i$
7       **if** $\operatorname{mse}(g, X) \leq \psi$ **then**             ▷ check whether $g$ vanishes
8         $\mathcal{G} \leftarrow \mathcal{G} \cup \{g\}$
9       **else**
10         $\mathcal{O} = \{t_1, \ldots, t_{\ell+1}\}_\sigma \leftarrow (\mathcal{O} \cup \{u_i\})_\sigma$
11       **end**
12     **end**
13     $d \leftarrow d + 1$
14 **end**

---

## 3   ORACLE APPROXIMATE VANISHING IDEAL ALGORITHM (OAVI)

In this section, we recall the oracle approximate vanishing ideal algorithm (OAVI) (Wirth & Pokutta, 2022) in Algorithm 1, a method for solving Problem 2.4.

### 3.1   ALGORITHM OVERVIEW

OAVI takes as input a data set set $X = \{\mathbf{x}_1, \ldots, \mathbf{x}_m\} \subseteq \mathbb{R}^n$, a vanishing parameter $\psi \geq 0$, and a tolerance $\tau \geq 2$ such that the constructed generators are $\tau$-bounded in the $\ell_1$-norm. From a high-level perspective, OAVI constructs a finite set of $(\psi, 1, \tau)$-approximately vanishing generators of the $(\psi, \tau)$-approximate vanishing ideal $\mathcal{I}_X^{\psi,\tau}$ by solving a series of constrained convex optimization problems. OAVI tracks the set of terms $\mathcal{O} \subseteq \mathcal{T}$ such that there does not exist a $(\psi, 1, \tau)$-approximately vanishing generator of $\mathcal{I}_X^{\psi,\tau}$ with terms only in $\mathcal{O}$ and the set of generators $\mathcal{G} \subseteq \mathcal{P}$ of $\mathcal{I}_X^{\psi,\tau}$. For every degree $d \geq 1$, OAVI computes the border $\partial_d \mathcal{O}$ in Line 2. Then, in Lines 3–12, for every term $u \in \partial_d \mathcal{O}$, OAVI determines whether there exists a $(\psi, 1, \tau)$-approximately vanishing generator $g$ of $\mathcal{I}_X^{\psi,\tau}$ with $\operatorname{lt}(g) = u$ and other terms only in $\mathcal{O}$ via oracle access to a solver of the constrained convex optimization problem in Line 5. If such a $g$ exists, it gets appended to $\mathcal{G}$ in Line 8. Otherwise, the term $u$ gets appended to $\mathcal{O}$ in Line 10. OAVI terminates when a degree $d \in \mathbb{N}$ is reached such that $\partial_d \mathcal{O} = \emptyset$. For ease of exposition, unless noted otherwise, we ignore that the optimization problem in Line 5 of OAVI is addressed only up to a certain accuracy $\epsilon > 0$. Throughout, we denote the output of OAVI by $\mathcal{G}$ and $\mathcal{O}$, that is, $(\mathcal{G}, \mathcal{O}) = \text{OAVI}(X, \psi, \tau)$. We replace the first letter in OAVI with the abbreviation corresponding to the solver used to solve the optimization problem in Line 5. For example, OAVI with solver CG is referred to as CGAVI.

### 3.2   PREPROCESSING WITH OAVI IN A CLASSIFICATION PIPELINE

OAVI can be used as a feature transformation method for a subsequently applied linear kernel support vector machine (SVM). Let $\mathcal{X} \subseteq \mathbb{R}^n$ and $\mathcal{Y} = \{1, \ldots, k\}$ be an input and output space, respectively. We are given a training sample $S = \{(\mathbf{x}_1, y_1), \ldots, (\mathbf{x}_m, y_m)\} \in (\mathcal{X} \times \mathcal{Y})^m$ drawn *i.i.d.* from some unknown distribution $\mathcal{D}$. Our task is to determine a *hypothesis* $h: \mathcal{X} \to \mathcal{Y}$ with small *generalization error* $\mathbb{P}_{(\mathbf{x},y) \sim \mathcal{D}}[h(\mathbf{x}) \neq y]$. For each class $i \in \{1, \ldots, k\}$, let $X^i = \{\mathbf{x}_j \in X \mid y_j = i\} \subseteq X = \{\mathbf{x}_1, \ldots, \mathbf{x}_m\}$ denote the subset of samples corresponding to class $i$ and construct a set of $(\psi, 1, \tau)$-approximately vanishing generators $\mathcal{G}^i$ of the $(\psi, \tau)$-approximate vanishing ideal $\mathcal{I}_{X^i}^{\psi,\tau}$ via OAVI, that is, $(\mathcal{G}^i, \mathcal{O}^i) = \text{OAVI}(X^i, \psi, \tau)$. We combine the sets of generators corresponding to the different classes to obtain $\mathcal{G} = \{g_1, \ldots, g_{|\mathcal{G}|}\} = \bigcup_{i=1}^k \mathcal{G}^i$, which encapsulates the feature

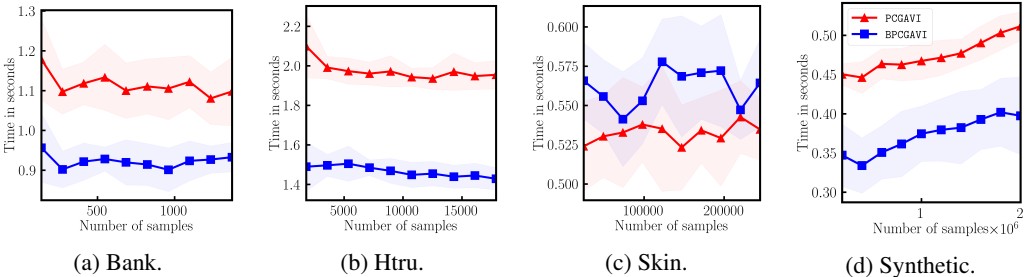

Figure 1: Training time comparisons with fixed $\psi = 0.005$, averaged over ten random runs with shaded standard deviations. On all data sets except skin, BPCGAVI is faster than PCGAVI.

transformation we are about to apply to data set $X$. As proposed in Livni et al. (2013), we transform the training sample $X$ via the feature transformation

$$\mathbf{x}_j \mapsto \tilde{\mathbf{x}}_j = \big(|g_1(\mathbf{x}_j)|, \ldots, |g_{|\mathcal{G}|}(\mathbf{x}_j)|\big)^\mathsf{T} \in \mathbb{R}^{|\mathcal{G}|} \tag{FT}$$

for $j = 1, \ldots, m$. The motivation behind this feature transformation is that a polynomial $g \in \mathcal{G}^i$ vanishes approximately over all $\mathbf{x} \in X^i$ and (hopefully) attains values that are far from zero over points $\mathbf{x} \in X \setminus X^i$. We then train a linear kernel SVM on the feature-transformed data $\tilde{S} = \{(\tilde{\mathbf{x}}_1, y_1), \ldots, (\tilde{\mathbf{x}}_m, y_m)\}$ with $\ell_1$-regularization to promote sparsity. If $\psi = 0$, $\tau = \infty$, and the underlying classes of $S$ belong to disjoint algebraic sets, then the different classes become linearly separable in the feature space corresponding to transformation (FT) and perfect classification accuracy is achieved with the linear kernel SVM (Livni et al., 2013).

### 3.3 SOLVING THE OPTIMIZATION PROBLEM IN LINE 5 OF OAVI

Setting $\tau = \infty$, the optimization problem in Line 5 becomes unconstrained and can, for example, be addressed with *accelerated gradient descent* (AGD) (Nesterov, 1983). If, however, $\tau < \infty$, OAVI satisfies two generalization bounds. Indeed, assuming that the data is of the form $X \subseteq [0, 1]^n$, the generators in $\mathcal{G}$ are guaranteed to also vanish on out-sample data. Furthermore, the combined approach of using generators obtained via OAVI to preprocess the data for a subsequently applied linear kernel SVM inherits the margin bound of the SVM (Wirth & Pokutta, 2022).

## 4 OAVI AT SCALE

Here, we address the main shortcoming of vanishing ideal algorithms: time, space, and evaluation complexities that depend superlinearly on $m$, where $m$ is the number of samples in the data set $X = \{\mathbf{x}_1, \ldots, \mathbf{x}_m\} \subseteq \mathbb{R}^n$. Recall the computational complexity of OAVI for $\tau = \infty$.

**Theorem 4.1** (Complexity (Wirth & Pokutta, 2022))**.** *Let $X = \{\mathbf{x}_1, \ldots, \mathbf{x}_m\} \subseteq \mathbb{R}^n$, $\psi \geq 0$, $\tau = \infty$, and $(\mathcal{G}, \mathcal{O}) = \text{OAVI}(X, \psi, \tau)$. Let $T_{\text{ORACLE}}$ and $S_{\text{ORACLE}}$ be the time and space complexities required to solve the convex optimization problem in Line 5 of OAVI, respectively. In the real number model, the time and space complexities of OAVI are $O((|\mathcal{G}| + |\mathcal{O}|)^2 + (|\mathcal{G}| + |\mathcal{O}|)T_{\text{ORACLE}})$ and $O((|\mathcal{G}| + |\mathcal{O}|)m + S_{\text{ORACLE}})$, respectively. The evaluation vectors of all polynomials in $\mathcal{G}$ over a set $Z = \{\mathbf{z}_1, \ldots, \mathbf{z}_q\} \subseteq \mathbb{R}^n$ can be computed in time $O((|\mathcal{G}| + |\mathcal{O}|)^2 q)$.*

Under mild assumptions, Wirth & Pokutta (2022) proved that $O(|\mathcal{G}| + |\mathcal{O}|) = O(mn)$, implying that OAVI's computational complexity is superlinear in $m$. In this work, we improve OAVI's computational complexity both with a tighter theoretical analysis and algorithmic modifications.

### 4.1 NUMBER-OF-SAMPLES-AGNOSTIC BOUND ON $|\mathcal{G}| + |\mathcal{O}|$

For $\psi > 0$, OAVI, ABM, AVI, and VCA construct approximately vanishing generators of the $(\psi, \tau)$-approximate vanishing ideal. Despite being designed for the approximate setting ($\psi > 0$), so far, the analysis of these algorithms was only conducted for the exact setting ($\psi = 0$). Below, we exploit that $\psi > 0$ in OAVI and obtain the first number-of-samples-agnostic bound on the size of a

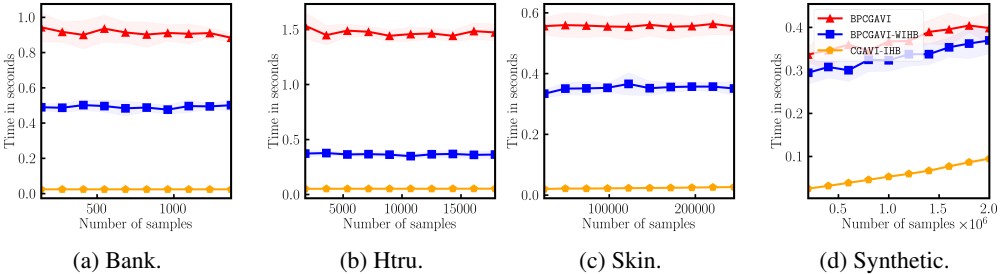

Figure 2: Training time comparisons with fixed $\psi = 0.005$, averaged over ten random runs with shaded standard deviations. `CGAVI-IHB` is faster than `BPCGAVI-WIHB`, which is faster than `BPCGAVI`.

generator-constructing algorithm's output. The result below also highlights the connection between the size of `OAVI`'s output $|\mathcal{G}| + |\mathcal{O}|$ and the extent of vanishing $\psi$: Large $\psi$ implies small $|\mathcal{G}| + |\mathcal{O}|$ and small $\psi$ implies large $|\mathcal{G}| + |\mathcal{O}|$.

**Theorem 4.2** (Bound on $|\mathcal{G}| + |\mathcal{O}|$). *Let* $X = \{\mathbf{x}_1, \ldots, \mathbf{x}_m\} \subseteq [0,1]^n$, $\psi \in \,]0,1[$, $D = \lceil -\log(\psi)/\log(4) \rceil$, $\tau \geq (3/2)^D$, *and* $(\mathcal{G}, \mathcal{O}) = \text{OAVI}(X, \psi, \tau)$. *Then,* `OAVI` *terminates after having constructed generators of degree D. Thus,* $|\mathcal{G}| + |\mathcal{O}| \leq \binom{D+n}{D}$.

The proof of Theorem 4.2 is presented in Appendix A. When $T_{\text{ORACLE}}$ and $S_{\text{ORACLE}}$ are linear in $m$ and polynomial in $|\mathcal{G}|+|\mathcal{O}|$, the time and space complexities of `OAVI` are linear in $m$ and polynomial in $n$. Furthermore, the evaluation complexity of `OAVI` is independent of $m$ and polynomial in $n$.

## 4.2 BLENDED PAIRWISE CONDITIONAL GRADIENTS ALGORITHM (BPCG)

Wirth & Pokutta (2022) solved the optimization problem in Line 5 of `OAVI` with the pairwise conditional gradients algorithm (`PCG`). In this section, we replace `PCG` with the blended pairwise conditional gradients algorithm (`BPCG`) (Tsuji et al., 2022), yielding an exponential improvement in time complexity of `OAVI` in the number of features $n$ in the data set $X = \{\mathbf{x}_1, \ldots, \mathbf{x}_m\} \subseteq [0,1]^n$.

Let $P \subseteq \mathbb{R}^\ell$, $\ell \in \mathbb{N}$, be a polytope of *diameter* $\delta > 0$ and *pyramidal width* $\omega > 0$, see Appendix D for details, and let $f: \mathbb{R}^\ell \to \mathbb{R}$ be a $\mu$-strongly convex and $L$-smooth function. `PCG` and `BPCG` address constrained convex optimization problems of the form

$$\mathbf{y}^* \in \text{argmin}_{\mathbf{y} \in P} f(\mathbf{y}). \tag{4.1}$$

With $\text{vert}(P)$ denoting the set of vertices of $P$, the convergence rate of `PCG` for solving (4.1) is $f(\mathbf{y}_T) - f(\mathbf{y}^*) \leq (f(\mathbf{y}_0) - f(\mathbf{y}^*))e^{-\rho_{\text{PCG}}T}$, where $\mathbf{y}_0 \in P$ is the starting point, $\mathbf{y}^*$ is the optimal solution to (4.1), and $\rho_{\text{PCG}} = \mu\omega^2/(4L\delta^2(3|\text{vert}(P)|! + 1))$ (Lacoste-Julien & Jaggi, 2015). The dependence of `PCG`'s time complexity on $|\text{vert}(P)|!$ is due to *swap steps*, which shift weight from the away to the Frank-Wolfe vertex but do not lead to a lot of progress. Since the problem dimension is $|\mathcal{O}|$, which can be of a similar order of magnitude as $|\mathcal{G}| + |\mathcal{O}|$, `PCG` can require a number of iterations that is exponential in $|\mathcal{G}| + |\mathcal{O}|$ to reach $\epsilon$-accuracy. `BPCG` does not perform swap steps and its convergence rate is $f(\mathbf{y}_T) - f(\mathbf{y}^*) \leq (f(\mathbf{y}_0) - f(\mathbf{y}^*))e^{-\rho_{\text{BPCG}}T}$, where $\rho_{\text{BPCG}} = 1/2 \min\{1/2, \mu\omega^2/(4L\delta^2)\}$ (Tsuji et al., 2022). Thus, to reach $\epsilon$-accuracy, `BPCG` requires a number of iterations that is only polynomial in $|\mathcal{G}| + |\mathcal{O}|$. For the optimization problem in Line 5 of `OAVI`, $A = \mathcal{O}(X) \in \mathbb{R}^{m \times \ell}$ and $\mu$ and $L$ are the smallest and largest eigenvalues of matrix $\frac{2}{m}A^\intercal A \in \mathbb{R}^{\ell \times \ell}$, respectively. Since the pyramidal width $\omega$ of the $\ell_1$-ball of radius $\tau - 1$ is at least $(\tau - 1)/\sqrt{\ell}$ (Lemma D.1), ignoring dependencies on $\mu$, $L$, $\epsilon$, and $\tau$, and updating $A^\intercal A$ and $A^\intercal \mathbf{b}$ as explained in the proof of Theorem C.1, the computational complexity of `BPCGAVI` (`OAVI` with solver `BPCG`) is as summarized below.

**Corollary 4.3** (Complexity). *Let* $X = \{\mathbf{x}_1, \ldots, \mathbf{x}_m\} \subseteq \mathbb{R}^n$, $\psi > 0$, $\tau \geq 2$, *and* $(\mathcal{G}, \mathcal{O}) = \text{BPCGAVI}(X, \psi, \tau)$. *In the real number model, the time and space complexities of* `BPCGAVI` *are* $O((|\mathcal{G}| + |\mathcal{O}|)^2 m + (|\mathcal{G}| + |\mathcal{O}|)^4)$ *and* $O((|\mathcal{G}| + |\mathcal{O}|)m + (|\mathcal{G}| + |\mathcal{O}|)^2)$, *respectively.*

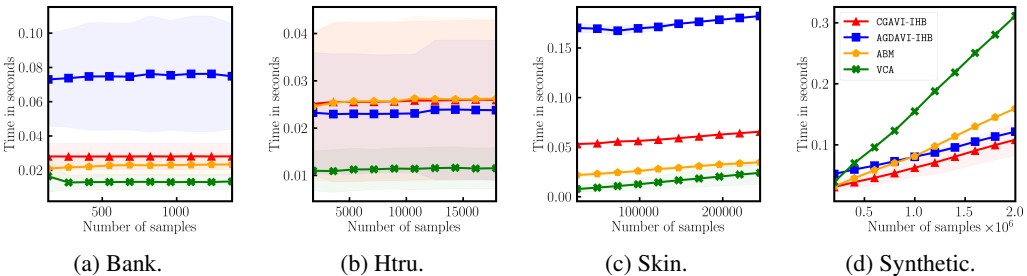

Figure 3: Training time comparisons, averaged over ten random runs with shaded standard deviations. For small data sets, `ABM` and `VCA` are faster than `OAVI`, but for synthetic, the training times of `ABM` and `VCA` scale worse than `OAVI`'s. See Appendix F.2 for details.

The results in Figure 1, see Appendix F.1 for details, show that replacing `PCG` with `BPCG` in `OAVI` often speeds up the training time of `OAVI`. For the data set skin in Figure 1c, see Table 1 for an overview of the data sets, it is not fully understood why `BPCGAVI` is slower than `PCGAVI`.

## 4.3 INVERSE HESSIAN BOOSTING (`IHB`)

We introduce inverse Hessian boosting (`IHB`) to speed up the training time of `OAVI` by multiple orders of magnitudes by exploiting the structure of the optimization problems solved in `OAVI`.

For ease of exposition, assume for now that $\tau = \infty$, in which case we would use `AGD` to solve the problem in Line 5 of `OAVI`. Letting $\ell = |\mathcal{O}|$, $A = \mathcal{O}(X) \in \mathbb{R}^{m \times \ell}$, $\mathbf{b} = u_i(X) \in \mathbb{R}^m$, and $f(\mathbf{y}) = \frac{1}{m}\|A\mathbf{y} + \mathbf{b}\|_2^2$, the optimization problem in Line 5 of `OAVI` takes the form $\mathbf{c} \in \operatorname{argmin}_{\mathbf{y} \in \mathbb{R}^\ell} f(\mathbf{y})$. Then, the gradient and Hessian of $f$ at $\mathbf{y} \in \mathbb{R}^\ell$ are $\nabla f(\mathbf{y}) = \frac{2}{m}A^\intercal(A\mathbf{y} + \mathbf{b}) \in \mathbb{R}^\ell$ and $\nabla^2 f(\mathbf{y}) = \frac{2}{m}A^\intercal A \in \mathbb{R}^{\ell \times \ell}$, respectively. By construction, the columns of $A = \mathcal{O}(X)$ are linearly independent. Hence, $A^\intercal A \in \mathbb{R}^{\ell \times \ell}$ is positive definite and invertible, $f$ is strongly convex, and the optimal solution $\mathbf{c}$ to the optimization problem in Line 5 of `OAVI` is unique. Further, for $\mathbf{y} \in \mathbb{R}^\ell$, we have $\nabla f(\mathbf{y}) = \mathbf{0}$ if and only if $\mathbf{y} = \mathbf{c}$. Instead of using `AGD` to construct an $\epsilon$-accurate solution to the optimization problem in Line 5 of `OAVI`, we could compute the optimal solution $\mathbf{c} = (A^\intercal A)^{-1}A^\intercal\mathbf{b} \in \mathbb{R}^\ell$. Since matrix inversions are numerically unstable, relying on them directly would make `OAVI` less robust, and approximately vanishing polynomials might not be correctly detected. Instead, we capitalize on the fact that the number of iterations of `AGD` to reach an $\epsilon$-minimizer depends on the Euclidean distance between the starting vector and the optimal solution. `IHB` refers to the procedure of passing $\mathbf{y}_0 = (A^\intercal A)^{-1}A^\intercal\mathbf{b} \in \mathbb{R}^\ell$ to `AGD` as a starting vector. Then, `AGD` often reaches $\epsilon$-accuracy in one iteration. In case $(A^\intercal A)^{-1}$ is not computed correctly due to floating-point errors, `AGD` still guarantees an $\epsilon$-accurate solution to the optimization problem. Thus, `IHB` can also be thought of as performing one iteration of *Newton's method* starting with iterate $\mathbf{0}$, see, for example, Galántai (2000), and passing the resulting vector as a starting iterate to `AGD`. We stress that the dimension of $A^\intercal A \in \mathbb{R}^{\ell \times \ell}$ is number-of-samples-agnostic, see Theorem 4.2.

`IHB` also works for $\tau < \infty$, in which case we use `CG` variants to address the optimization problem in Line 5 of `OAVI` that takes the form $\mathbf{c} \in \operatorname{argmin}_{\mathbf{y} \in P} f(\mathbf{y})$, where $P = \{\mathbf{y} \in \mathbb{R}^\ell \mid \|\mathbf{y}\|_1 \leq \tau - 1\}$. In the problematic case

$$\|\mathbf{y}_0\|_1 = \|(A^\intercal A)^{-1}A^\intercal\mathbf{b}\|_1 > \tau - 1, \tag{INF}$$

polynomial $g = \sum_{j=1}^\ell c_j t_j + u_i$ constructed in Line 6 of `OAVI` might not vanish approximately even though there exists $h \in \mathcal{P}$ with $\operatorname{lt}(h) = u_i$, $\operatorname{ltc}(h) = 1$, non-leading terms only in $\mathcal{O}$, and $\|h\|_1 > \tau$ that vanishes exactly over $X$. Thus, $\operatorname{mse}(h, X) = 0 \leq \psi < \operatorname{mse}(g, X)$ and `OAVI` does not append $g$ to $\mathcal{G}$ and instead updates $\mathcal{O} \leftarrow (\mathcal{O} \cup \{u_i\})_\sigma$ and $A \leftarrow (A, \mathbf{b}) \in \mathbb{R}^{m \times (\ell+1)}$. Since $\operatorname{mse}(h, X) = 0$, we just appended a linearly dependent column to $A$, making $A$ rank-deficient, $(A^\intercal A)^{-1}$ ill-defined, and `IHB` no longer applicable. To address this problem, we could select $\tau$ adaptively, but this would invalidate the learning guarantees of `OAVI` that rely on $\tau$ being a constant. In practice, we fix $\tau \geq 2$ and stop using `IHB` as soon as (INF) holds for the first time, preserving the generalization bounds of `OAVI`. Through careful updates of $(A^\intercal A)^{-1}$, the discussion in Appendix C.1 implies the following complexity result for `CGAVI-IHB` (`CGAVI` with `IHB`).

Table 1: Overview of data sets. All data sets are binary classification data sets and are retrieved from the UCI Machine Learning Repository (Dua & Graff, 2017) and additional references are provided.

| Data set | Full name | # of samples | # of features |
|---|---|---|---|
| bank | banknote authentication | 1,372 | 4 |
| credit | default of credit cards (Yeh & Lien, 2009) | 30,000 | 22 |
| htru | HTRU2 (Lyon et al., 2016) | 17,898 | 8 |
| skin | skin (Bhatt & Dhall, 2010) | 245,057 | 3 |
| spam | spambase | 4,601 | 57 |
| synthetic | synthetic, see Appendix E | 2,000,000 | 3 |

**Corollary 4.4** (Complexity). *Let $X = \{\mathbf{x}_1, \ldots, \mathbf{x}_m\} \subseteq \mathbb{R}^n$, $\psi \geq 0$, $\tau \geq 2$ large enough such that (INF) never holds, and $(\mathcal{G}, \mathcal{O}) = $ CGAVI-IHB$(X, \psi, \tau)$. Assuming CG terminates after a constant number of iterations due to IHB, the time and space complexities of CGAVI-IHB are $O((|\mathcal{G}| + |\mathcal{O}|)^2 m + (|\mathcal{G}| + |\mathcal{O}|)^3)$ and $O((|\mathcal{G}| + |\mathcal{O}|)m + (|\mathcal{G}| + |\mathcal{O}|)^2)$, respectively.*

In Appendix C.2, we introduce *weak inverse Hessian boosting* (WIHB), a variant of IHB that speeds up CGAVI variants while preserving sparsity-inducing properties. In Figure 2, see Appendix F.1 for details, we observe that CGAVI-IHB is faster than BPCGAVI-WIHB, which is faster than BPCGAVI. In Figure 3, see Appendix F.2 for details, we compare the training times of OAVI-IHB to ABM and VCA. In Figure 3d, we observe that OAVI-IHB's training time scales better than ABM's and VCA's.

## 5 NUMERICAL EXPERIMENTS

Unless noted otherwise, the setup for the numerical experiments applies to all experiments in the paper. Experiments are implemented in PYTHON and performed on an Nvidia GeForce RTX 3080 GPU with 10GB RAM and an Intel Core i7 11700K 8x CPU at 3.60GHz with 64 GB RAM. Our code is publicly available on GitHub.

We implement OAVI as in Algorithm 1 with convex solvers CG, PCG, BPCG, and AGD and refer to the resulting algorithms as CGAVI, PCGAVI, BPCGAVI, and AGDAVI, respectively. Solvers are run for up to 10,000 iterations. For the CG variants, we set $\tau = 1,000$. The CG variants are run up to accuracy $\epsilon = 0.01 \cdot \psi$ and terminated early when less than $0.0001 \cdot \psi$ progress is made in the difference between function values, when the coefficient vector of a generator is constructed, or if we have a guarantee that no coefficient vector of a generator can be constructed. AGD is terminated early if less than $0.0001 \cdot \psi$ progress is made in the difference between function values for 20 iterations in a row or the coefficient vector of a generator is constructed. OAVI implemented with WIHB or IHB is referred to as OAVI-WIHB or OAVI-IHB, respectively. We implement ABM as in Limbeck (2013) but instead of applying the *singular value decomposition* (SVD) to the matrix corresponding to $A = \mathcal{O}(X)$ in OAVI, we apply the SVD to $A^\intercal A$ in case this leads to a faster training time and we employ the border as in Definition 2.5. We implement VCA as in Livni et al. (2013) but instead of applying the SVD to the matrix corresponding to $A = \mathcal{O}(X)$ in OAVI, we apply the SVD to $A^\intercal A$ in case this leads to a faster training time. We implement a polynomial kernel SVM with a one-versus-rest approach using the SCIKIT-LEARN software package (Pedregosa et al., 2011) and run the polynomial kernel SVM with $\ell_2$-regularization up to tolerance $10^{-3}$ or for up to 10,000 iterations.

We preprocess with OAVI, ABM, and VCA for a linear kernel SVM as in Section 3.2 and refer to the combined approaches as OAVI*, ABM*, and VCA*, respectively. The linear kernel SVM is implemented using the SCIKIT-LEARN software package and run with $\ell_1$-penalized squared hinge loss up to tolerance $10^{-4}$ or for up to 10,000 iterations. For OAVI*, ABM*, and VCA*, the hyperparameters are the vanishing tolerance $\psi$ and the $\ell_1$-regularization coefficient of the linear kernel SVM. For the polynomial kernel SVM, the hyperparameters are the degree and the $\ell_2$-regularization coefficient. Table 1 and 3 contain overviews of the data sets and hyperparameter values, respectively.

### 5.1 EXPERIMENT: PERFORMANCE

We compare the performance of CGAVI-IHB*, BPCGAVI-WIHB*, AGDAVI-IHB*, ABM*, VCA*, and polynomial kernel SVM on the data sets credit, htru, skin, and spam.

Table 2: Numerical results averaged over ten random $60\%/40\%$ train/test partitions with best results in bold. For approaches other than BPCGAVI-WIHB*, spar$(\mathcal{G}) < 0.01$ and we omit the results.

| Algorithms | | Data sets | | | |
|---|---|---|---|---|---|
| | | credit | htru | skin | spam |
| Error | CGAVI-IHB* | 18.08 | 2.09 | 0.23 | 6.69 |
| | AGDAVI-IHB* | 18.08 | 2.09 | 0.23 | 6.69 |
| | BPCGAVI-WIHB* | **17.98** | **2.05** | **0.22** | 6.71 |
| | ABM* | 18.36 | 2.09 | 0.43 | **6.67** |
| | VCA* | 19.85 | 2.11 | 0.24 | 7.15 |
| | SVM | 18.34 | 2.08 | 2.25 | 7.13 |
| Time | CGAVI-IHB* | $1.3 \times 10^2$ | $2.3 \times 10^1$ | $1.0 \times 10^2$ | $8.3 \times 10^1$ |
| | AGDAVI-IHB* | $1.9 \times 10^2$ | $2.8 \times 10^1$ | $1.1 \times 10^2$ | $3.1 \times 10^2$ |
| | BPCGAVI-WIHB* | $3.7 \times 10^3$ | $8.0 \times 10^2$ | $5.6 \times 10^2$ | $4.2 \times 10^2$ |
| | ABM* | $1.2 \times 10^2$ | $2.4 \times 10^1$ | $6. \times 10^1$ | $1.7 \times 10^2$ |
| | VCA* | $\mathbf{2.4 \times 10^1}$ | $6.2 \times 10^0$ | $\mathbf{1.4 \times 10^1}$ | $6.5 \times 10^1$ |
| | SVM | $8.9 \times 10^1$ | $\mathbf{4.1 \times 10^0}$ | $7.1 \times 10^2$ | $\mathbf{2.2 \times 10^0}$ |
| $\|\mathcal{G}\| + \|\mathcal{O}\|$ | CGAVI-IHB* | 82.40 | 27.90 | 39.00 | 786.60 |
| | AGDAVI-IHB* | 82.40 | 27.90 | 39.00 | 786.60 |
| | BPCGAVI-WIHB* | 106.40 | 55.20 | 39.00 | 653.00 |
| | ABM* | 51.40 | 28.70 | 19.30 | **261.00** |
| | VCA* | **49.80** | **19.00** | **12.40** | 1766.40 |
| (SPAR) | BPCGAVI-WIHB* | **0.67** | **0.52** | **0.03** | **0.71** |

**Setup.** We tune the hyperparameters on the training data using threefold cross-validation. We retrain on the entire training data set using the best combination of hyperparameters and evaluate the classification error on the test set and the hyperparameter optimization time. For the generator-constructing methods, we also compare $|\mathcal{G}| + |\mathcal{O}|$, where $|\mathcal{G}| = \sum_i |\mathcal{G}^i|$, $|\mathcal{O}| = \sum_i |\mathcal{O}|^i$, and $(\mathcal{G}^i, \mathcal{O}^i)$ is the output of applying a generator-constructing algorithm to samples belonging to class $i$ and the *sparsity of the feature transformation* $\mathcal{G} = \bigcup_i \mathcal{G}^i$, which is defined as

$$\text{spar}(\mathcal{G}) = (\sum_{g \in \mathcal{G}} g_z)/(\sum_{g \in \mathcal{G}} g_e) \in [0, 1], \tag{SPAR}$$

where for a polynomial $g = \sum_{j=1}^{k} c_j t_j + t$ with $\text{lt}(g) = t$, $g_e = k$ and $g_z = |\{c_j = 0 \mid j \in \{1, \ldots, k\}\}|$, that is, $g_e$ and $g_z$ are the number of non-leading and the number of zero coefficient vector entries of $g$, respectively. Results are averaged over ten random $60\%/40\%$ train/test partitions.

**Results.** The results are presented in Table 2. OAVI* admits excellent test-set classification accuracy. BPCGAVI-WIHB*, in particular, admits the best test-set classification error on all data sets but one. Hyperparameter tuning for OAVI* is often slightly slower than for ABM* and VCA*. Since BPCGAVI-WIHB* does not employ IHB, the approach is always slower than CGAVI-IHB* and AGDAVI-IHB*. For all data sets but credit and skin, the hyperparameter optimization time for the SVM is shorter than for the other approaches. On skin, since the training time of the polynomial kernel SVM is superlinear in $m$, the hyperparameter training for the polynomial kernel SVM is slower than for the other approaches. For data sets with few features $n$, the magnitude of $|\mathcal{G}| + |\mathcal{O}|$ is often the smallest for VCA*. However, for spam, a data set with $n = 57$, as already pointed out by Kera & Hasegawa (2019) as the spurious vanishing problem, we observe VCA's tendency to create unnecessary generators. Finally, only BPCGAVI-WIHB* constructs sparse feature transformations, potentially explaining the excellent test-set classification accuracy of the approach.

In conclusion, the numerical results justify considering OAVI as the generator-constructing algorithm of choice, as it is the only approximate vanishing ideal algorithm with known learning guarantees and, in practice, performs better than or similar to related approaches.

ACKNOWLEDGEMENTS

We would like to thank Gabor Braun for providing us with the main arguments for the proof of the lower bound on the pyramidal width of the $\ell_1$-ball. This research was partially funded by the Deutsche Forschungsgemeinschaft (DFG, German Research Foundation) under Germany´s Excellence Strategy – The Berlin Mathematics Research Center MATH$^+$ (EXC-2046/1, project ID 390685689, BMS Stipend) and JST, ACT-X Grant Number JPMJAX200F, Japan.

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

## A  Missing proofs

*Proof of Theorem 4.2.* Let $X = \{\mathbf{x}_1, \ldots, \mathbf{x}_m\} \subseteq [0,1]^n$ and let $t_1, \ldots, t_n \in \mathcal{T}_1$ be the degree-1 monomials. Suppose that during OAVI's execution, for some degree $d \in \mathbb{N}$, OAVI checks whether the term $u = \prod_{j=1}^n t_j^{\alpha_j} \in \partial_d \mathcal{O}$, where $\sum_{j=1}^n \alpha_j = d$ and $\alpha_j \in \mathbb{N}$ for all $j \in \{1, \ldots, n\}$, is the leading term of a $(\psi, 1, \tau)$-approximately vanishing generator with non-leading terms only in $\mathcal{O}$. Let

$$h = \prod_{j=1}^n \left( t_j - \frac{1}{2}\mathbb{1} \right)^{\alpha_j}, \tag{A.1}$$

where $\mathbb{1}$ denotes the constant-1 monomial, that is, $\mathbb{1}(\mathbf{x}) = 1$ for all $\mathbf{x} \in \mathbb{R}^n$. Note that $\|h\|_1 \leq \left(\frac{3}{2}\right)^d \leq \tau$, guaranteeing that $h$ can be constructed in Lines 5–6 of OAVI. Since $u \in \partial_d \mathcal{O}$, for any term $t \in \mathcal{T}$ such that $t \mid u$ and $t \neq u$, it holds that $t \in \mathcal{O}$ and thus, $h$ is a polynomial with $\mathrm{ltc}(h) = 1$, $\mathrm{lt}(h) = u$, and other terms only in $\mathcal{O}$. Under the assumption that the convex optimization oracle in OAVI is accurate, $\mathrm{mse}(h, X) \geq \mathrm{mse}(g, X)$, where $g$ is the polynomial constructed during Lines 5 and 6 of OAVI. Hence, proving that $h$ vanishes approximately implies that $g$ vanishes approximately. Note that for all $\mathbf{x} \in X$ and $t \in \mathcal{T}_1$, it holds that $t(\mathbf{x}) \in [0,1]$, and, thus, $|t(\mathbf{x}) - \frac{1}{2}| \leq \frac{1}{2}$. We obtain

$$\mathrm{mse}(g, X) \leq \mathrm{mse}(h, X)$$
$$= \frac{1}{m}\|h(X)\|_2^2$$
$$= \frac{1}{m} \sum_{\mathbf{x} \in X} \left( \prod_{j=1}^n \left( t_j(\mathbf{x}) - \frac{1}{2}\mathbb{1}(\mathbf{x}) \right)^{\alpha_j} \right)^2$$
$$\leq \max_{\mathbf{x} \in X} \prod_{j=1}^n \left| t_j(\mathbf{x}) - \frac{1}{2} \right|^{2\alpha_j}$$
$$\leq 4^{-d}.$$

Since $\mathrm{mse}(g, X) \leq 4^{-d} \leq \psi$ is satisfied for $d \geq D := \left\lceil -\frac{\log(\psi)}{\log(4)} \right\rceil$, OAVI terminates after reaching degree $D$. Thus, at the end of OAVI's execution, $|\mathcal{G}| + |\mathcal{O}| \leq \binom{D+n}{D} \leq (D+n)^D$. $\qquad\square$

## B  Blended pairwise conditional gradients algorithm (BPCG)

The blended pairwise conditional gradients algorithm (BPCG) (Tsuji et al., 2022) is presented in Algorithm 2.

## C  Additional information on IHB

We discuss the computational complexity of IHB in Appendix C.1 and WIHB in Appendix C.2.

### C.1  Computational complexity of IHB

The main cost of IHB is the inversion of the matrix $A^\intercal A \in \mathbb{R}^{\ell \times \ell}$, which generally requires $O(\ell^3)$ elementary operations. Since OAVI solves a series of quadratic convex optimization problems that differ from each other only slightly, we can maintain and update $(A^\intercal A)^{-1}$ using $O(\ell^2)$ instead of $O(\ell^3)$ elementary operations.

**Theorem C.1** (IHB update cost). *Let $A \in \mathbb{R}^{m \times \ell}$, $A^\intercal A \in \mathbb{R}^{\ell \times \ell}$, $(A^\intercal A)^{-1}$, and $\mathbf{b} \in \mathbb{R}^\ell$ be given. In case $\|\mathbf{b}\|_2 > 0$ and $\mathbf{b}^\intercal A(A^\intercal A)^{-1} A^\intercal \mathbf{b} \neq \|\mathbf{b}\|_2^2$,*

$$\tilde{A} = (A, \mathbf{b}) \in \mathbb{R}^{m \times (\ell+1)}, \qquad \tilde{A}^\intercal \tilde{A} \in \mathbb{R}^{(\ell+1) \times (\ell+1)}, \qquad (\tilde{A}^\intercal \tilde{A})^{-1} \in \mathbb{R}^{(\ell+1) \times (\ell+1)} \tag{C.1}$$

*can be constructed in $O(\ell m + \ell^2)$ elementary operations.*

---

**Algorithm 2:** Blended pairwise conditional gradients algorithm (BPCG)

---

**Input** : A smooth and convex function $f$, a starting vertex $\mathbf{y}_0 \in \text{vert}(P)$.
**Output:** A point $\mathbf{y}_T \in P$.

---

1   $S^{(0)} \leftarrow \{\mathbf{y}_0\}$

2   $\lambda_{\mathbf{y}_0}^{(0)} \leftarrow 1$

3   $\lambda_{\mathbf{y}_0}^{(0)} \leftarrow 0$ for $\boldsymbol{v} \in \text{vert}(P) \setminus \{\mathbf{y}_0\}$

4   **for** $t = 0, \dots, T-1$ **do**

5      $\mathbf{a}_t \in \text{argmax}_{\boldsymbol{v} \in S^{(t)}} \langle \nabla f(\mathbf{y}_t), \boldsymbol{v} \rangle$                    ▷ away vertex

6      $\mathbf{q}_t \in \text{argmin}_{\boldsymbol{v} \in S^{(t)}} \langle \nabla f(\mathbf{y}_t), \boldsymbol{v} \rangle$               ▷ local FW vertex

7      $\mathbf{w}_t \in \text{argmin}_{\boldsymbol{v} \in \text{vert}(P)} \langle \nabla f(\mathbf{y}_t), \boldsymbol{v} \rangle$           ▷ FW vertex

8      **if** $\langle \nabla f(\mathbf{y}_t), \mathbf{w}_t - \mathbf{y}_t \rangle \geq \langle \nabla f(\mathbf{y}_t), \mathbf{q}_t - \mathbf{a}_t \rangle$ **then**

9          $\mathbf{d}_t \leftarrow \mathbf{q}_t - \mathbf{a}_t$

10         $\gamma_t \in \text{argmin}_{\gamma \in [0, \lambda_{\mathbf{a}_t}^{(t)}]} f(\mathbf{y}_t + \gamma \mathbf{d}_t)$

11         $\lambda_{\boldsymbol{v}}^{(t+1)} \leftarrow \lambda_{\boldsymbol{v}}^{(t)}$ for $\boldsymbol{v} \in \text{vert}(P) \setminus \{\mathbf{a}_t, \mathbf{q}_t\}$

12         $\lambda_{\mathbf{a}_t}^{(t+1)} \leftarrow \lambda_{\mathbf{a}_t}^{(t)} - \gamma_t$

13         $\lambda_{\mathbf{q}_t}^{(t+1)} \leftarrow \lambda_{\mathbf{q}_t}^{(t)} + \gamma_t$

14      **else**

15         $\mathbf{d}_t \leftarrow \mathbf{w}_t - \mathbf{y}_t$

16         $\gamma_t \in \text{argmin}_{\gamma \in [0,1]} f(\mathbf{y}_t + \gamma \mathbf{d}_t)$

17         $\lambda_{\boldsymbol{v}}^{(t+1)} \leftarrow (1 - \gamma_t)\lambda_{\boldsymbol{v}}^{(t)}$ for $\boldsymbol{v} \in \text{vert}(P) \setminus \{\mathbf{w}_t\}$

18         $\lambda_{\mathbf{w}_t}^{(t+1)} \leftarrow (1 - \gamma_t)\lambda_{\mathbf{w}_t}^{(t)} + \gamma_t$

19      **end**

20      $S^{(t+1)} \leftarrow \{\boldsymbol{v} \in \text{vert}(P) \mid \lambda_{\boldsymbol{v}}^{(t+1)} > 0\}$

21      $\mathbf{y}_{t+1} \leftarrow \mathbf{y}_t + \gamma_t \mathbf{d}_t$

22   **end**

---

*Proof.* Let $B = A^\intercal A \in \mathbb{R}^{\ell \times \ell}$, $\tilde{B} = \tilde{A}^\intercal \tilde{A} \in \mathbb{R}^{(\ell+1) \times (\ell+1)}$, $N = B^{-1} \in \mathbb{R}^{\ell \times \ell}$, and $\tilde{N} = \tilde{B}^{-1} = (\tilde{A}^\intercal \tilde{A})^{-1} \in \mathbb{R}^{(\ell+1) \times (\ell+1)}$. In $O(m\ell)$ elementary operations, we can construct $A^\intercal \mathbf{b} \in \mathbb{R}^\ell$, $\mathbf{b}^\intercal \mathbf{b} = \|\mathbf{b}\|_2^2 \in \mathbb{R}$, and $\tilde{A} = (A, \mathbf{b}) \in \mathbb{R}^{m \times (\ell+1)}$ and in additional $O(\ell^2)$ elementary operations, we can construct

$$\tilde{B} = \tilde{A}^\intercal \tilde{A} = \begin{pmatrix} B & A^\intercal \mathbf{b} \\ \mathbf{b}^\intercal A & \|\mathbf{b}\|_2^2 \end{pmatrix} \in \mathbb{R}^{(\ell+1) \times (\ell+1)}.$$

We can then compute $\tilde{N} = \tilde{B}^{-1} = (\tilde{A}^\intercal \tilde{A})^{-1} \in \mathbb{R}^{(\ell+1) \times (\ell+1)}$ in additional $O(\ell^2)$ elementary operations. We write

$$\tilde{N} = \begin{pmatrix} \tilde{N}_1 & \tilde{\mathbf{n}}_2 \\ \tilde{\mathbf{n}}_2^\intercal & \tilde{n}_3 \end{pmatrix} \in \mathbb{R}^{(\ell+1) \times (\ell+1)},$$

where $\tilde{N}_1 \in \mathbb{R}^{\ell \times \ell}$, $\tilde{\mathbf{n}}_2 \in \mathbb{R}^\ell$, and $\tilde{n}_3 \in \mathbb{R}$. Then, it has to hold that

$$\tilde{B}\tilde{N} = \begin{pmatrix} B & A^\intercal \mathbf{b} \\ \mathbf{b}^\intercal A & \|\mathbf{b}\|_2^2 \end{pmatrix} \begin{pmatrix} \tilde{N}_1 & \tilde{\mathbf{n}}_2 \\ \tilde{\mathbf{n}}_2^\intercal & \tilde{n}_3 \end{pmatrix} = I \in \mathbb{R}^{(\ell+1) \times (\ell+1)},$$

where $I$ is the identity matrix. Note that $\mathbf{b}^\intercal A \tilde{\mathbf{n}}_2 + \|\mathbf{b}\|_2^2 \tilde{n}_3 = 1$. Thus,

$$\tilde{n}_3 = \frac{1 - \mathbf{b}^\intercal A \tilde{\mathbf{n}}_2}{\|\mathbf{b}\|_2^2} \in \mathbb{R}, \tag{C.2}$$

which is well-defined due to the assumption $\|\mathbf{b}\|_2 > 0$. Since $\mathbf{b}^\intercal A$ is already computed, once $\tilde{\mathbf{n}}_2$ is computed, the computation of $\tilde{n}_3$ requires only additional $O(\ell)$ elementary operations. Similarly, we have that

$$B\tilde{\mathbf{n}}_2 + A^\intercal \mathbf{b} \tilde{n}_3 = \mathbf{0} \in \mathbb{R}^\ell. \tag{C.3}$$

Plugging (C.2) into (C.3), we obtain $(B - \frac{A^\intercal \mathbf{b}\mathbf{b}^\intercal A}{\|\mathbf{b}\|_2^2})\tilde{\mathbf{n}}_2 = -\frac{A^\intercal \mathbf{b}}{\|\mathbf{b}\|_2^2}$. Thus,

$$\tilde{\mathbf{n}}_2 = -\left(B - \frac{A^\intercal \mathbf{b}\mathbf{b}^\intercal A}{\|\mathbf{b}\|_2^2}\right)^{-1}\frac{A^\intercal \mathbf{b}}{\|\mathbf{b}\|_2^2}.$$

The existence of $(B - \frac{A^\intercal \mathbf{b}\mathbf{b}^\intercal A}{\|\mathbf{b}\|_2^2})^{-1}$ follows from the Sherman-Morrison formula (Sherman & Morrison, 1950; Bartlett, 1951) and the assumption that $\mathbf{b}^\intercal A(A^\intercal A)^{-1}A^\intercal \mathbf{b} \neq \|\mathbf{b}\|_2^2$. Then, again using the Sherman-Morrison formula,

$$\tilde{\mathbf{n}}_2 = -\left(B^{-1} + \frac{B^{-1}A^\intercal \mathbf{b}\mathbf{b}^\intercal A B^{-1}}{\|\mathbf{b}\|_2^2 - \mathbf{b}^\intercal A B^{-1} A^\intercal \mathbf{b}}\right)\frac{A^\intercal \mathbf{b}}{\|\mathbf{b}\|_2^2},$$

which can be computed using additional $O(\ell^2)$ elementary operations. Finally, we construct $\tilde{N}_1$, which is determined by $B\tilde{N}_1 + A^\intercal \mathbf{b}\tilde{\mathbf{n}}_2^\intercal = I \in \mathbb{R}^{\ell\times\ell}$. Thus, $\tilde{N}_1 = B^{-1} - B^{-1}A^\intercal \mathbf{b}\tilde{\mathbf{n}}_2^\intercal \in \mathbb{R}^{\ell\times\ell}$, which can be computed in additional $O(\ell^2)$ elementary operations since $B^{-1}A^\intercal \mathbf{b} \in \mathbb{R}^\ell$ is already computed. In summary, we require $O(\ell m + \ell^2)$ elementary operations. Note that even if we do not compute $(\tilde{A}^\intercal \tilde{A})^{-1}$, we still require $O(\ell m + \ell^2)$ elementary operations. $\qquad\square$

In the remark below, we discuss the literature related to IHB.

**Remark C.2** (Work related to IHB). The proof of Theorem C.1 is similar to the proof that the inverse of the Hessian can be updated efficiently in the *online Newton algorithm* (Hazan et al., 2007). Both proofs rely on the Sherman-Morrison formula (Sherman & Morrison, 1950; Bartlett, 1951). However, in our setting, the updates occur column-wise instead of row-wise. Updating $(A^\intercal A)^{-1}$ column-wise for generator-constructing algorithms was already discussed in Limbeck (2013) using QR decompositions without addressing the numerical instability of inverse matrix updates.

Next, we prove that Theorem C.1 implies the improved computational complexity of CGAVI-IHB in Corollary 4.4.

*Proof of Corollary 4.4.* Suppose that CGAVI-IHB is currently executing Lines 5–11 of Algorithm 1 for a particular term $u_i \in \partial_d\mathcal{O}$ and recall the associated notations $\ell = |\mathcal{O}|$, $A = \mathcal{O}(X) \in \mathbb{R}^{m\times\ell}$, $\mathbf{b} = u_i(X)$, $f(\mathbf{y}) = \frac{1}{m}\|A\mathbf{y} + \mathbf{b}\|_2^2$, and $P = \{\mathbf{y} \in \mathbb{R}^\ell \mid \|\mathbf{y}\|_1 \leq \tau - 1\}$. Then, the optimization problem in Line 5 of Algorithm 1 takes the form

$$\mathbf{c} \in \arg\min_{\mathbf{y}\in P} f(\mathbf{y}).$$

We first prove that the violation of any of the two assumptions of Theorem C.1 implies the existence of a $(\psi, 1, \tau)$-approximately vanishing generator $g$. We prove this claim by treating the two assumptions separately:

1. If $\|\mathbf{b}\|_2 = 0$, it holds that $\mathrm{mse}(u_i, X) = 0$. By the assumption that $\tau \geq 2$ is large enough to guarantee that (INF) never holds, CGAVI-IHB constructs a $(\psi, 1, \tau)$-approximately vanishing generator in Line 6.

2. If $\mathbf{b}^\intercal A(A^\intercal A)^{-1}A^\intercal \mathbf{b} = \|\mathbf{b}\|_2^2$, it holds that $f(-(A^\intercal A)^{-1}A^\intercal \mathbf{b}) = 0$. Thus, by the assumption that $\tau \geq 2$ is large enough to guarantee that (INF) never holds, CGAVI-IHB constructs a $(\psi, 1, \tau)$-approximately vanishing generator in Line 6.

Since an update of $(A^\intercal A)^{-1}$ is necessary only if Line 10 is executed, that is, when there does not exist a $(\psi, 1, \tau)$-approximately vanishing generator $g$, we never have to update $(A^\intercal A)^{-1}$ when the assumptions of Theorem C.1 are violated. Thus, by Theorem C.1, we can always update $A$, $A^\intercal A$, and $(\tilde{A}^\intercal \tilde{A})^{-1}$ in $O(\ell m + \ell^2) \leq O((|\mathcal{G}| + |\mathcal{O}|)m + (|\mathcal{G}| + |\mathcal{O}|)^2)$ elementary operations using space $O(\ell m + \ell^2) \leq O((|\mathcal{G}| + |\mathcal{O}|)m + (|\mathcal{G}| + |\mathcal{O}|)^2)$. Then, since we run CG for a constant number of iterations, the time and space complexities of Lines 5–11 are $T_{\mathrm{ORACLE}} = O((|\mathcal{G}| + |\mathcal{O}|)m + (|\mathcal{G}| + |\mathcal{O}|)^2)$ and $S_{\mathrm{ORACLE}} = O((|\mathcal{G}| + |\mathcal{O}|)m + (|\mathcal{G}| + |\mathcal{O}|)^2)$, respectively. $\qquad\square$

Note that the time and space complexities in Corollary 4.4 also hold for AGDAVI-IHB with $\tau = \infty$.

---

**Algorithm 3:** Blended pairwise conditional gradients approximate vanishing ideal algorithm with weak inverse Hessian boosting (`BPCGAVI-WIHB`)

---

**Input** : A data set $X = \{\mathbf{x}_1, \ldots, \mathbf{x}_m\} \subseteq \mathbb{R}^n$ and parameters $\psi \geq \epsilon \geq 0$ and $\tau \geq 2$.
**Output:** A set of polynomials $\mathcal{G} \subseteq \mathcal{P}$ and a set of monomials $\mathcal{O} \subseteq \mathcal{T}$.

---

1   $d \leftarrow 1, \mathcal{O} = \{t_1\}_\sigma \leftarrow \{\mathbb{1}\}_\sigma, \mathcal{G} \leftarrow \emptyset$
2   **while** $\partial_d \mathcal{O} = \{u_1, \ldots, u_k\}_\sigma \neq \emptyset$ **do**
3      **for** $i = 1, \ldots, k$ **do**
4          $\ell \leftarrow |\mathcal{O}| \in \mathbb{N}, A \leftarrow \mathcal{O}(X) \in \mathbb{R}^{m \times \ell}, \mathbf{b} \leftarrow u_i(X) \in \mathbb{R}^m, P = \{\mathbf{y} \in \mathbb{R}^\ell \mid \|\mathbf{y}\|_1 \leq \tau - 1\}$
5          $\mathbf{y}_0 \leftarrow (A^\intercal A)^{-1} A^\intercal \mathbf{b} \in \mathbb{R}^\ell$
6          Solve $\mathbf{c} \in \operatorname{argmin}_{\mathbf{y} \in P} \frac{1}{m}\|A\mathbf{y} + \mathbf{b}\|_2^2$ up to tolerance $\epsilon$ using `CG` with starting vector $\mathbf{y}_0$.
7          $g \leftarrow \sum_{j=1}^\ell c_j t_j + u_i$            ▷ a non-sparse polynomial
8          **if** $\operatorname{mse}(g, X) \leq \psi$ **then**        ▷ check whether $g$ vanishes
9              Solve $\mathbf{d} \in \operatorname{argmin}_{\mathbf{y} \in P} \frac{1}{m}\|A\mathbf{y} + \mathbf{b}\|_2^2$ up to tolerance $\epsilon$ using `BPCG` and a vertex of $P$ as starting vector.
10              $h \leftarrow \sum_{j=1}^\ell d_j t_j + u_i$         ▷ a sparse polynomial
11              **if** $\operatorname{mse}(h, X) \leq \psi$ **then**       ▷ check whether $h$ vanishes
12                 $\mathcal{G} \leftarrow \mathcal{G} \cup \{h\}$
13              **else**
14                 $\mathcal{G} \leftarrow \mathcal{G} \cup \{g\}$
15              **end**
16          **else**
17              $\mathcal{O} = \{t_1, \ldots, t_{\ell+1}\}_\sigma \leftarrow (\mathcal{O} \cup \{u_i\})_\sigma$
18          **end**
19      **end**
20      $d \leftarrow d + 1$
21 **end**

---

## C.2   Weak inverse Hessian boosting (`WIHB`)

So far, we have introduced `IHB` to drastically speed-up training of `AGDAVI` and `CGAVI`. A drawback of using `CGAVI-IHB` is that the initialization of `CG` variants with a non-sparse initial vector such as $\mathbf{y}_0 = (A^\intercal A)^{-1} A^\intercal \mathbf{b}$ leads to the construction of generally non-sparse generators. In this section, we explain how to combine the speed-up of `IHB` with the sparsity-inducing properties of `CG` variants, referring to the resulting technique as weak inverse Hessian boosting (`WIHB`). Specifically, we present `WIHB` with `BPCGAVI` (`OAVI` with solver `BPCG`), referred to as `BPCGAVI-WIHB` in Algorithm 3.

The high-level idea of `BPCGAVI-WIHB` is to use `IHB` and vanilla `CG` to quickly check whether a $(\psi, 1, \tau)$-approximately vanishing generator exists. If it does, we then use `BPCG` to try and construct a $(\psi, 1, \tau)$-approximately vanishing generator that is also sparse.

We proceed by giving a detailed overview of `BPCGAVI-WIHB`. In Line 5 of `BPCGAVI-WIHB`, we construct $\mathbf{y}_0$. Then, in Line 6, we solve $\mathbf{c} \in \operatorname{argmin}_{\mathbf{y} \in P} \frac{1}{m}\|A\mathbf{y} + \mathbf{b}\|_2^2$ up to tolerance $\epsilon$ using `CG` with starting vector $\mathbf{y}_0 = (A^\intercal A)^{-1} A^\intercal \mathbf{b}$. Since $\mathbf{y}_0$ is a well-educated guess for the solution to the constrained optimization problem $\operatorname{argmin}_{\mathbf{y} \in P} \frac{1}{m}\|A\mathbf{y} + \mathbf{b}\|_2^2$ in Line 6, `CG` often runs for only very few iterations. The drawback of using the non-sparse $\mathbf{y}_0$ as a starting vector is that $\mathbf{c}$ constructed in Line 6 and $g$ constructed in Line 7 are generally non-sparse. We alleviate the issue of non-sparsity of $g$ in Lines 8–18. In Line 8, we first check whether $g$ is a $(\psi, 1, \tau)$-approximately vanishing generator of $X$. If $g$ does not vanish approximately, we know that there does not exist an approximately vanishing generator with leading term $u_i$ and we append $u_i$ to $\mathcal{O}$ in Line 17. If, however, $\operatorname{mse}(g, X) \leq \psi$, we solve the constrained convex optimization problem in Line 6 again in Line 9 up to tolerance $\epsilon$ using `BPCG` and a vertex of the $\ell_1$-ball $P$ as starting vector. This has two consequences:

     1. The vector $\mathbf{d}$ constructed in Line 9 tends to be sparse, as corroborated by the results in Table 2.

2. The execution of Line 9 tends to take longer than the execution of Line 6 since BPCG's starting vector is not necessarily close in Euclidean distance to the optimal solution.

Then, in Line 10, we construct the polynomial $h$, which tends to be sparse. If $h$ is a $(\psi, 1, \tau)$-approximately vanishing generator, we append the sparse $h$ to $\mathcal{G}$ in Line 12. If it happens that $\mathrm{mse}(g, X) \leq \psi < \mathrm{mse}(h, X)$, we append the non-sparse $g$ to $\mathcal{G}$ in Line 14. Following the discussion of Section 4.3, should (INF) ever hold, that is, $\|\mathbf{y}_0\|_1 > \tau - 1$, we can no longer proceed with BPCGAVI-WIHB as we can no longer guarantee that the inverse of $A^\intercal A$ exists. In that case, we proceed with vanilla BPCGAVI for all terms remaining in the current and upcoming borders.

The discussion above illustrates that BPCGAVI-WIHB solves $\mathrm{argmin}_{\mathbf{y} \in P} \frac{1}{m} \|A\mathbf{y} + \mathbf{b}\|_2^2$ with BPCG $|\mathcal{G}|$ times as opposed to BPCGAVI, which solves $\mathrm{argmin}_{\mathbf{y} \in P} \frac{1}{m} \|A\mathbf{y} + \mathbf{b}\|_2^2$ with BPCG $|\mathcal{G}| + |\mathcal{O}| - 1$ times. Thus, BPCGAVI-WIHB combines the sparsity-inducing properties of BPCG with the speed-up of IHB without any of IHB's drawbacks. The speed-up of BPCGAVI-WIHB compared to BPCGAVI is evident from the numerical experiments in Figure 2 and the results of Table 2 indicate that the sparsity-inducing properties of BPCG are successfully exploited.

# D  PYRAMIDAL WIDTH OF THE $\ell_1$-BALL

Pena & Rodriguez (2019) showed that the pyramidal width $\omega$ of the $\ell_1$-ball of radius $\tau$, that is, $P = \{\mathbf{x} \in \mathbb{R}^\ell \mid \|\mathbf{x}\|_1 \leq \tau\} \subseteq \mathbb{R}^\ell$ is given by

$$\omega = \min_{F \in \mathrm{faces}\,(P), \emptyset \subsetneq F \subsetneq P} \mathrm{dist}\,(F, \mathrm{conv}(\mathrm{vert}(P) \setminus F)),$$

where, for a polytope $P \subseteq \mathbb{R}^\ell$, $\mathrm{faces}\,(P)$ denotes the set of faces of $P$ and $\mathrm{vert}(P)$ denotes the set of vertices of $P$, for a set $F \subseteq \mathbb{R}^\ell$, $\mathrm{conv}(F)$ is the convex hull of $F$, and for two disjoint sets $F, G \subseteq \mathbb{R}^\ell$, $\mathrm{dist}\,(F, G)$ is the Euclidean distance between $F$ and $G$.

**Lemma D.1** (Pyramidal width of the $\ell_1$-ball). *The pyramidal width of the $\ell_1$-ball of radius $\tau$, that is, $P = \{\mathbf{x} \in \mathbb{R}^\ell \mid \|\mathbf{x}\|_1 \leq \tau\} \subseteq \mathbb{R}^\ell$, is lower bounded by $\omega \geq \frac{\tau}{\sqrt{\ell-1}}$.*

*Proof.* Let $\mathbf{e}^{(i)} \in \mathbb{R}^\ell$ denote the $i$th unit vector. Note that a non-trivial face of $P = \mathrm{conv}\{\pm\tau\mathbf{e}^{(1)}, \ldots, \pm\tau\mathbf{e}^{(n)}\} \subseteq \mathbb{R}^\ell$ that is not $P$ itself cannot contain both $\tau\mathbf{e}^{(i)}$ and $-\tau\mathbf{e}^{(i)}$ for any $i \in \{1, \ldots, \ell\}$. Thus, due to symmetry, we can assume that any non-trivial face of $P$ that is not $P$ itself is of the form $F = \mathrm{conv}(\{\tau\mathbf{e}^{(1)}, \ldots, \tau\mathbf{e}^{(k)}\})$ for some $k \in \{1, \ldots, \ell - 1\}$. Then,

$$G := \mathrm{conv}(\mathrm{vert}(P) \setminus F) = \mathrm{conv}(\{-\tau\mathbf{e}^{(1)}, \ldots, -\tau\mathbf{e}^{(k)}, \pm\tau\mathbf{e}^{(k+1)}, \ldots, \pm\tau\mathbf{e}^{(\ell)}\}).$$

We have that

$$\mathrm{dist}\,(F, G) = \min_{\mathbf{u} \in F, \boldsymbol{v} \in G} \|u - v\|_2$$

$$= \min_{\mathbf{u} \in F, \boldsymbol{v} \in G} \sqrt{\sum_{i=1}^{k}(u_i - v_i)^2 + \sum_{j=k+1}^{\ell} v_j^2}$$

$$\geq \min_{\mathbf{u} \in F, \boldsymbol{v} \in G} \sqrt{\sum_{i=1}^{k}(u_i - v_i)^2} \qquad \triangleright \text{ since } v_j^2 \geq 0 \text{ for all } j \in \{k+1, \ldots, \ell\}$$

$$\geq \min_{\mathbf{u} \in F} \sqrt{\sum_{i=1}^{k} u_i^2} \qquad \triangleright \text{ since } v_i \leq 0 \leq u_i \text{ for all } i \in \{1, \ldots, k\}$$

$$= \sqrt{\frac{\tau^2}{k}}.$$

Since $\sqrt{\frac{\tau^2}{k}}$ is minimized for $k \in \{1, \ldots, \ell - 1\}$ as large as possible, it follows that $\omega \geq \frac{\tau}{\sqrt{\ell-1}}$. $\qquad \square$

Table 3: Hyperparameter ranges for numerical experiments.

| Hyperparameters | Values |
|---|---:|
| $\psi$ | 0.1, 0.05, 0.01, 0.005, 0.001, 0.0005, 0.0001 |
| Regularization coefficient for SVMs | 0.1, 1, 10 |
| Degree for polynomial kernel SVM | 1, 2, 3, 4 |

## E  THE SYNTHETIC DATA SET

The synthetic data set consists of three features and contains two different classes. The samples $\mathbf{x}$ that belong to the first class are generated such that they satisfy the equation

$$x_1^2 + 0.01x_2 + x_3^2 - 1 = 0$$

and samples that belong to the second class are generated such that they satisfy the equation

$$x_1^2 + x_3^2 - 1.3 = 0.$$

The samples are perturbed with additive Gaussian noise with mean 0 and standard deviation 0.05.

## F  ADDITIONAL NUMERICAL EXPERIMENTS

In this section, we provide details for the numerical experiments conducted in Figures 1, 2, and 3.

### F.1  EXPERIMENT: SPEEDING UP CGAVI

We compare the training times of PCGAVI, BPCGAVI, BPCGAVI-WIHB, and CGAVI-IHB on the data sets bank, htru, skin, and synthetic.

**Setup.**  For a single run, we randomly split the data set into subsets of varying sizes. Then, for fixed $\psi = 0.005$, we run the generator-constructing algorithms on subsets of the full data set of varying sizes and plot the training times, which are the times required to run PCGAVI, BPCGAVI, BPCGAVI-WIHB, and CGAVI-IHB once for each class. The results are averaged over ten random runs and standard deviations are shaded in Figures 1 and 2.

**Results.**  The results are presented in Figures 1 and 2. In Figure 1, we observe that the training times for BPCGAVI are often shorter than for PCGAVI, except for the skin data set. In Figure 2, we observe that IHB not only leads to theoretically better training times but also speeds up training in practice: CGAVI-IHB is always faster than BPCGAVI. Furthermore, WIHB is indeed the best of both worlds: BPCGAVI-WIHB is always faster than BPCGAVI but preserves the sparsity-inducing properties of BPCG. The latter can also be seen in Table 2, where BPCGAVI-WIHB* is the only algorithmic approach that constructs sparse generators.

### F.2  EXPERIMENT: SCALABILITY COMPARISON

We compare the training times of CGAVI-IHB, AGD-IHB, ABM, and VCA on the data sets bank, htru, skin, and synthetic.

**Setup.**  For a single run, on at most 10,000 samples of the data set, we tune the hyperparameters of generator-constructing algorithms OAVI, ABM, and VCA and a subsequently applied linear kernel SVM using threefold cross-validation. Then, using the determined hyperparameters, we run only the generator-constructing algorithms on subsets of the full data set of varying sizes and plot the training times, which are the times required to run CGAVI-IHB, AGD-IHB, ABM, and VCA once for each class. The results are averaged over ten random runs and standard deviations are shaded in Figure 3.

**Results.**   The results are presented in Figure 3. For small data sets, we observe that `ABM` and `VCA` are faster than `OAVI`. However, when the number of samples in the data set is large, as in the synthetic data set, `OAVI` can be trained faster than `ABM` and `VCA`. `AGDAVI-IHB` is slower than `CGAVI-IHB` because `AGD` cannot use the Frank-Wolfe gap as an early termination criterion to quickly decide that a solution close enough to the optimum is reached.

