# OpenReview forum: "Approximate Vanishing Ideal Computations at Scale"
_ICLR.cc/2023/Conference — ICLR 2023 poster_

### Official Review · Reviewer_thxT · 2022-10-24

**Confidence:** 3
**Correctness:** 4
**Technical Novelty And Significance:** 3
**Empirical Novelty And Significance:** 3
**Recommendation:** 8

**Clarity, Quality, Novelty And Reproducibility:**

The paper is novel, cites the relevant research and shows how it extends the current state of the art.
It is generally well written and easy to follow, although a stronger integration between the main part of the paper and the appendix would be preferable (see weaknesses above)

**Strength And Weaknesses:**

Strengths:
* VCA is an intuitive and promising concept that suffers from some limitations in practice. This paper makes an important contribution to approximate versions thereof (namely OAVI) in that it shows that the complexity is actually linear in the number of samples (and not superlinear as believed hitherto)
* The paper further proposes practical improvements (PCG and BPCG) and inverse Hessian boosting for the optimization steps involved.
* The contributions are well-explained and comprehensible and related to prior work

Weaknesses:
* The empirical evaluation shows a slight benefit of the proposed method; yet, for many experiments, the advantages in comparison to the classical VCA algorithm are less clear. My experiences were that VCA suffers particularly from sample-complexity whenever higher order polynomials are required (because of the large amount of generators that might be generated). This alligns well with the experiments in Figure 3 and suggests that the benefit of the proposed method becomes relevant only for more complex problems. I am afraid though that the experiments performed are not sufficient for making a particularly strong point for the OAVI type algorithms. I think that including more complex datasets (similar to spam) would really highlight the benefits.
* The proof of Theorem 4.2 (a crucial piece of the paper) relies on the assumption that the oracle in OAVI is accurate. This seems to be a relative hand waving argument and -- at least for me -- it is not obvious if or under which conditions this is the case
* The paper has a strong theoretical flavor. Yet, the most important contributions are completely hidden in the appendix. At the very least it would be desirable to have a brief summary of the proofs in the main part of the paper and to link to the respective proofs. For example, found it unnecessary tedious to look through the whole appendix for the proof of Th 4.2 without any pointer (only to finally find it under "missing proofs").

**Summary Of The Paper:**

This paper extends the recent spur of research on VCA. In particular, it provides theoretical properties on the computational complexity of OAVI by utilizing the fact that the ideals must only vanish approximately and not exactly and improve the training time by clever tricks in the optimization.

**Summary Of The Review:**

The paper tackles a relevant problem and makes progress both from a theoretical and from a practical perspective. The main insights could be better highlighted in the paper though.
Moreover, the experiments are well-designed and correct but focus on too many 'easy' problems that do not reveal the capabilities of the proposed approaches well enough.

---

> ### Author Response · Authors · 2022-11-13
> **Response**
>
> Dear Reviewer,
>
> We would like to thank you for taking the time to read our paper, providing valuable feedback, and that you consider our work's readability a strength, an aspect we focused on during the writing process.
>
> Point 1:
> In the experiments, we focus on testing OAVI on various data sets (different number of samples and features) to give a fair comparison with other methods. As you pointed out, for some data sets the performance of OAVI is comparable to VCA. We do, however, believe that these results are enough to justify using OAVI on the presented data sets as the method of choice because OAVI has stronger theoretical guarantees than other generator-constructing algorithms (Wirth & Pokutta, 2022). First, OAVI is the only generator-constructing algorithm that admits learning guarantees when tau =/= infinity. For OAVI, constructed generators are guaranteed to vanish on out-sample data and the combined approach of using OAVI to transform features for a subsequently applied linear kernel SVM inherits the generalization bound of the SVM. Second, solving the convex subproblems in OAVI with CG adds another benefit to OAVI in that the constructed generators are now also sparse. The only real shortcoming of OAVI was that the algorithm requires more time to train than ABM and VCA. This shortcoming was also the main motivator of this paper, and we believe that IHB and WIHB address this issue to a large extent. Now, OAVI is a fast and theoretically well-supported alternative to ABM and VCA.
> We appreciate your careful observation that OAVI may show better performance on complex data sets. After reading your comment, we performed some more experiments to corroborate this statement and found that as the number of features grows large (50 +), the size of VCA's output tends to blow up. However, the main focus of this paper is not to highlight VCA's shortcomings. To keep the message of our work clear, "scaling of approximate vanishing ideal algorithms and OAVI in particular", we believe that adding more data sets similar to spam would not contribute to the core message. Indeed, focusing on VCA's shortcomings might even be distracting from the main points. If you do, however, insist that adding these experiments to the paper would greatly improve our work, we are happy to follow through and put them in the final version.
>
> Point 2:
> We next discuss your comments regarding Theorem 4.2. We indeed ignore the fact that the convex solvers are only returning and epsilon-accurate solution. The convex optimization oracle can never really be assumed to be accurate. In Theorem 4.2, we left out solver inaccuracies not because they would complicate the proof but because they make the exposition more complicated. Even assuming the convex solver returns an epsilon-accurate solution, which, e.g., CG does, Theorem 4.2 still holds except that any occurrence of psi has to be replaced by psi - epsilon. We will add a small note on this in the final version of the paper.
>
> Point 3:
> We will make sure to add a pointer to the proof of Theorem 4.2 in the final version. The theorem is the main contribution and its proof should be easy to find. Ideally, we would have put the proof in the main body, but, unfortunately, we are limited by the available space.

---

> > ### Comment · Reviewer_thxT · 2022-11-22
> > **Response**
> >
> > Thanks for your response to my points. I firmly believe that adding a small note regarding the effect epsilon-accurate solutions and incorporating Th. 4.2. better will improve the paper.
> > As you mentioned in your response, the aim is to discuss the theoretical properties of OAVI. Thus, I agree that it might not be necessary to add additional experiments and that this might distract from the main point. Still, I think that a brief discussion of what we discussed here can help the reader to better understand some of the practical differences between VCA and OAVI.

---

> > > ### Author Response · Authors · 2022-11-30
> > > **Response**
> > >
> > > Dear Reviewer,
> > >
> > > We will make sure to do that.
> > >
> > > Best,
> > > the authors

---

> > > > ### Comment · Reviewer_thxT · 2022-12-08
> > > > **Response**
> > > >
> > > > Given that the first response of the authors effectively addressed two of the main points for criticism and the authors promised to address the final weakness by brief discussion on the differences between VCA and OAVI, I am willing to increase my score accordingly.
> > > > I strongly encourage the authors to take the above discussion into account and revise the paper accordingly in case of acceptance.

---

### Official Review · Reviewer_qggK · 2022-10-24

**Confidence:** 3
**Correctness:** 4
**Technical Novelty And Significance:** 3
**Empirical Novelty And Significance:** 3
**Recommendation:** 6

**Clarity, Quality, Novelty And Reproducibility:**

The paper is written clearly. I believe enough details are given about the proposed algorithm, as well as the algorithm the authors are building on.

**Strength And Weaknesses:**

The algorithm seems to require a lot of hand-holding and might be hard to tune in practice. For example, the numerical instabilities that come with matrix inversion need to be handled carefully, as pointed out by the authors.

Experimental results seem impressive in general, but it would be good to understand why the method suffers in one particular dataset.

Experiments in section 5 do not seem to report runtime results unless I am missing them.

**Summary Of The Paper:**

The authors propose a modified version of the OAVI algorithm for computing the generators of the set of polynomials who has a given dataset as roots - called the ideal. They show theoretically that their proposed modified algorithm yields significant savings in computational complexity. They then use these ideals to solve a classification task and show good test set performance.

**Summary Of The Review:**

I am not an expert in this field and my limited understanding based on reading the paper is that it seems to be a worthy publication given the experimental results.

After rebuttal:

Thank you for your responses and clarifications. I strongly suggest the authors to write their modified algorithm explicitly in a separate algorithmic environment. After a closer look, I found the first paragraph of page 15 (in appendix) to appreciate the steps they had to take to accommodate for the change they did to line 5's optimization problem. It will greatly help the readers to compare with the existing work. Since there is no option of 7 as a score, and I cannot justify an 8 in terms of novelty and impact (no comparison with other classifiers in the experiments), I will keep my score of 6 (weak accept).

---

> ### Author Response · Authors · 2022-11-13
> **Response**
>
> Dear Reviewer,
>
> We would like to thank you for taking the time to read our paper and for the feedback provided.
>
> Below, we address your comments:
>
> Point 1:
> In practice, OAVI requires surprisingly little tuning. The only parameter that we tuned is psi, which can be done with a simple grid search or Bayesian optimization. Preliminary experiments illustrated that tuning epsilon, tau, and stopping criteria of CG have almost no effect on OAVI's performance. Thus, tuning OAVI is similar to tuning ABM and VCA, fast and simple.
>
> Point 2:
> Generally, we agree that matrix inversions are tricky and not very stable. In this work, we only use matrix inversions to obtain a good starting vector for our convex solvers. Since our solvers are run to (almost) optimality irrespective of the starting vector, we enjoy the speed-up of matrix inversion without suffering the consequences of the potential numerical instability. In practice, we observe that the starting vector provided via matrix inversion tends to lead to a massive speed-up, see Figure 2.
>
> Point 3:
> We are not sure which data set you are referring to since OAVI performs well on all data sets. We are happy to provide an explanation if you specify which data set and performance measure you are referring to.
>
> Point 4:
> Table 2 currently only shows the hyperparameter optimization time, which was a conscious decision on our part. The numerical results of Wirth & Pokutta (2022) already provide the full story of test times: The test times of OAVI, ABM, AVI, and VCA are all similarly fast and do not scale with increasing number of samples, whereas the test time of the SVM is known to increases with increasing number of samples. To avoid unnecessary cluttering of the experiment section, we omitted these results, but if you believe that they could provide value to the reader, we are happy to put them back into the final version of the paper.

---

### Official Review · Reviewer_298L · 2022-10-27

**Confidence:** 3
**Correctness:** 4
**Technical Novelty And Significance:** 3
**Empirical Novelty And Significance:** Not applicable
**Recommendation:** 8

**Clarity, Quality, Novelty And Reproducibility:**

Please see "Weaknesses" above.

------- AFTER THE REBUTTAL PHASE -----

In their rebuttal, the authors very clearly explained both their algorithm and the intuition for *what*'s being done differently compared to the prior work of Wirth and Pokutta. I strongly suggest incorporating into the final version this explanation since it significantly improves the paper in my opinion. With this added explanation, I believe the paper is quite well-written.

**Strength And Weaknesses:**

Strength: The scalability of an existing algorithm for an important problem

Weakness:

1. I am not sure how standard the notation used in the paper is (I wasn't familiar with it), and the description in the Preliminaries section was not enough for me to clearly follow the details. For instance, the notion of degree-lexicographical term ordering could be motivated better; I found the notation too succinct for my taste. I think this could be rectified by providing a working example for the reader. Along the same lines, since the paper builds heavily upon the OAVI algorithm and its notation, it might be useful to the reader to better motivate all the key concepts therein.

2. The paper could provide some intuition for the proof of Theorem 4.2, which shows the runtime improvement: What exactly is the analysis doing differently from that of Wirth and Pokutta? What step, exactly, makes the prior work's analysis sub-optimal? I believe answering these questions would make the paper more insightful and the concepts more broadly applicable.

------ AFTER THE REBUTTAL PHASE ----

The above weaknesses have been adequately addressed.

**Summary Of The Paper:**

The key contribution of the paper is providing a faster algorithm (with improved dimension dependence) for computing generators of the approximate vanishing ideal of a set of points. To this end, it provides (1) a novel analysis of an existing algorithm for this purpose ("OAVI" by Wirth and Pokutta), and (2) a simple modification to accelerate OAVI. The notion of vanishing ideals is used as a core technique in several classical machine learning tasks like feature selection and transformation. Efficient algorithms for this subroutine are, therefore, a problem of clear interest to ICLR.



**Summary Of The Review:**

My preliminary review is, unfortunately, to reject the paper: I believe the problem studied is important, and I am ready to believe the (theoretical) claims made, but I think there is plenty of room for improvement in the clarity. I am, however, happy to raise my score and change my opinion if the authors can submit a rebuttal clarifying notation and providing intuition for the main proof.

----- AFTER THE REBUTTAL PHASE ---

I am convinced of the importance of the paper's contributions and vote to accept the paper. I do strongly recommend that the authors incorporate into this version the following (1) a plain description (in words) of what the steps in the algorithm are doing, and (2) intuition for why their new methods improve upon the prior analysis (i.e., the explanations they provided in the rebuttal). Doing so would significantly increase readability.

---

> ### Author Response · Authors · 2022-11-13
> **Response**
>
> Dear Reviewer,
>
> Thank you for taking the time to read our paper and provide helpful suggestions.
>
> Point 1:
> We aimed to make the paper as accessible as possible, and we thus take your comment regarding readability very seriously and try to address it below.
>
> Since our paper does not introduce OAVI but improves the algorithm and its analysis, we decided to keep the preliminaries as succinct as possible. As it turns out, we wrote too succinctly, and we will motivate the notation more in the final version of the paper. For now, we hope that the explanations below can shed more light on OAVI.
>
> First and foremost, OAVI is an algorithm that constructs polynomials that vanish (psi, 1, tau)-approximately over a given data set X, i.e., OAVI constructs generators of the approximate vanishing ideal of X. These generators are of interest because they capture polynomial structure in the data and can be used to map the data set into a higher dimensional feature space in which different classes can become linearly separable (See also Section 3.2). A detailed description of OAVI is presented in Section 3.1, and here we focus on the most important concepts. Throughout its execution, OAVI keeps track of two sets: O, the set of monomials such that there does not exist a (psi, 1, tau)-approximately vanishing generator with terms only in O, and G, the set of (psi, 1, tau)-approximately vanishing generators that OAVI already constructed. For every monomial encountered, OAVI checks in Lines 4 to 11 whether the monomial is a leading term of a (psi, 1, tau)-approximately vanishing generator or whether the term belongs to O. Then, OAVI accordingly updates O or G and proceeds to the next term. For degree d, OAVI does not have to check all monomials of degree d but only those that are contained in the border (Definition 2.5): If there exists a generator g with leading term t in G, then no u such that u is divisible by t has to be checked anymore.
>
> Example: Suppose that polynomial x_2 + x_1 is added to G. This polynomial has leading term x_2. Then, during the computations corresponding to degree 3, OAVI would no longer have to check whether monomials x_2x_1 and x_2^2 are leading terms of (psi, 1, tau)-approximately vanishing generators.
>
> Thus, incorporating the border prevents OAVI from having to check too many monomials (i.e, solving optimization problems too many times) and drastically speeds up the algorithm. Our numerical experiments highlight this. In Table 2, for the spam data set, we can see that OAVI and ABM, which both incorporate the border as in Definition 2.5, construct a significantly more compact output (|G| + |O|) than VCA, an algorithm which cannot employ the border.
>
> Next, we explain in which order OAVI checks the terms of a specific degree d. This order is determined by the term ordering sigma, which is necessary for any monomial-aware algorithm. Any term ordering that first sorts monomials degree-wise can be used with OAVI. For readability, we decided to present the algorithm with the degree-lexicographical term ordering to not distract the reader with the complex definition of term ordering that provides little insight into how OAVI operates. Different term orderings will lead to different outputs for OAVI, but comparing different term orderings remains unexplored in the literature.
> If you specify more points that we could elaborate on, we are happy to do so.
>
> Point 2:
> Currently, the intuition behind the proof of Theorem 4.2 is discussed at the bottom of page 5. In the final version, we will make sure to provide further details. For now, we hope that the paragraph below can provide the desired insights:
> OAVI, ABM, AVI, and VCA are algorithms that construct generators for approximate vanishing ideals; that is, they consider the setting psi > 0. Despite constructing approximately vanishing generators (psi > 0), so far, the analysis of these algorithms was only done for exactly vanishing generators (psi = 0). In the proof of Theorem 4.2, we, for the first time, exploit that psi > 0 (as opposed to performing the analysis for psi = 0). Our analysis provides a number-of-samples-agnostic bound on |G| + |O|; that is, we prove that the size of the output of OAVI does not depend on the number of samples in the data set X. Further, our result highlights the connection between the extent of vanishing, psi, and the size of OAVI's output: increasing psi leads to smaller |G| + |O|, and increasing psi leads to larger |G| + |O|. This relation between psi and |G| + |O| has been known empirically for various algorithms (e.g. ABM, VCA) but not explained theoretically.
>
> We consider Theorem 4.2 to be significant because (i) it allows us to analyze the behavior of a generator construction algorithm in the approximate setting (psi > 0) for the first time, and (ii) it elucidates the sample-size-agnosticity and the effects of psi on |G| + |O|, which have been suggested only empirically but never explained theoretically in prior studies.

---

> ### Author Response · Authors · 2022-11-17
> **Response II**
>
> Dear Reviewer,
>
> We believe we have addressed all the concerns in your review. If you do not have any further issues, we would kindly ask you to reconsider your score.
>
> Best,
> the authors

---

### Decision · Program_Chairs · 2023-01-20

**Decision:**

Accept: poster

**Justification For Why Not Higher Score:**

The experiments seem a bit lacking and the problem a bit niche.

**Justification For Why Not Lower Score:**

Direct theoretical improvement over a previous work on this topic.

**Metareview: Summary, Strengths And Weaknesses:**

The problem was viewed as a little bit niche, but there is recent work on this in AISTATS that this submission seems to directly improve upon. In particular, they take the same algorithm in that paper and do non-trivial technical work to modify a line of it and show approximate convergence and hence and improved time complexity. The rebuttal was convincing regarding this aspect. The reviewers were not really convinced by the experiments as there seem to maybe be better ways to do the tasks they considered (e.g., deeper models), but the theory did seem interesting enough.

**Note From Pc:**

if the above contains the word "oral" or "spotlight" please see: "oral" presentation means -> notable-top-5% and "spotlight" means -> notable-top-25%. As stated in our emails, we are disassociating presentation type from AC recommendations

**Summary Of Ac-Reviewer Meeting:**

We mainly discussed the technical differences between this paper and the earlier paper of Wirth and Pokutta.